# The Effect of Postinduction Blood Glucose on Intraoperative Hypothermia

**DOI:** 10.3390/medicina59020395

**Published:** 2023-02-17

**Authors:** Zhangtian Shen, Kosuke Kuroda, Hiroshi Morimatsu

**Affiliations:** Department of Anesthesiology and Resuscitology, Okayama University Graduate School of Medicine, Dentistry and Pharmaceutical Sciences, 2-5-1 Shikata-cho, Kitaku, Okayama 700-8558, Japan

**Keywords:** blood glucose, hemoglobin A1c, hypothermia, thermoregulation, laparoscopy, type 2 diabetes

## Abstract

*Background and Objectives*: Hypothermia frequently occurs in patients undergoing surgery and is associated with adverse complications. Therefore, this study aimed to investigate the postinduction blood glucose and occurrence of intraoperative hypothermia in patients undergoing laparoscopic surgery. *Materials and Methods*: This retrospective observational study included 334 patients aged ≥20 years who had undergone elective laparoscopic surgery. The primary outcome of this study was the incidence of intraoperative hypothermia. Stratified analysis revealed differences between patients with and without diabetes. *Results*: Hypothermia occurred in 200 (59.9%) patients. In multivariate analysis, out-of-range postinduction glucose was independently associated with hypothermia (>150 mg/dL: odds ratio 2.17, 95% confidence interval (1.02, 4.61), *p* = 0.045; <110 mg/dL: odds ratio 2.02, 95% confidence interval (1.15, 3.55), *p* = 0.015), whereas preoperative HbA1c >6% was not significantly associated with hypothermia (odds ratio 1.02, 95% confidence interval (0.56, 1.84), *p* = 0.961). Considering only patients with diabetes, the incidence of hypothermia was lower (*p* = 0.002), the duration of hypothermia was shorter (*p* = 0.007), and the minimum temperature was higher (*p* = 0.006) in those with a postinduction glucose level of 110–150 mg/dL. *Conclusions*: The postinduction glucose level is independently associated with intraoperative hypothermia. Out-of-range postinduction glucose appeared to have an impact on the development of hypothermia in patients with diabetes, especially those with a postinduction glucose level <110 mg/dL.

## 1. Introduction

Hypothermia, defined as a core body temperature <36 °C, is one of the most common adverse events in patients undergoing surgery. Impaired thermoregulation and heat redistribution due to general anesthesia and direct heat loss from surgery are notable causes of intraoperative hypothermia. Even mild hypothermia can cause adverse complications, such as surgical site infections [1], cardiovascular adverse events [2], blood loss [3], and prolongation of postanesthetic recovery [4]. Methods to prevent perioperative hypothermia include various experimental warming interventions, forced air warming, infusion warming, and thermal suits are currently recommended to maintain perioperative body temperature in daily practice [5,6,7,8,9]. However, hypothermia still occurs in patients, even when active warming systems are applied.

Clinically, short- and long-term glucose control is also associated with postoperative outcomes [10,11]. Although, current evidence shows that poor glycemic control can lead to peripheral neuropathy and microvascular dysfunction [12], the mechanism of pathophysiology is very complex. A commonly used explanation is oxidative stress [13]. Inflammatory cytokines have the main control in these processes because of their inflammatory promoting activity [14,15], which, in turn, may be related to intraoperative hypothermia [16,17]. However, the mechanism of how blood glucose affects thermoregulation remains unclear. Further information regarding this may be gleaned from studies in patients with diabetes. Due to diabetic autonomic neuropathy and diabetic microvascular disease, these patients appear to have an impaired vascular responsiveness to cold [18], resulting in a reduced ability to maintain their core temperature during cold exposure [19]. Therefore, active temperature management and moderate preoperative glycemic control could be important to enhance patient safety and satisfaction.

Few studies have reported on intraoperative hypothermia between patients with and without diabetes [20]. However, to our knowledge, no studies have examined how perioperative glycemic control affects intraoperative hypothermia in laparoscopic surgery. Therefore, this study aimed to investigate the blood glucose levels and occurrence of intraoperative hypothermia in patients undergoing laparoscopic surgery. We hypothesized that patients with poorly controlled postinduction blood glucose levels are at higher risk for the development of intraoperative hypothermia.

## 2. Materials and Methods

### 2.1. Study Design

This was a single-center, retrospective, and observational investigation approved by the Institutional Review Board of the Okayama University Graduate School of Medicine, Dentistry, and Pharmaceutical Sciences (reference number 2106-037). This study was performed in accordance with the ethical standards laid down in the Declaration of Helsinki and its later amendments.

### 2.2. Patients and Data Collection

We conducted a retrospective observational study, including all patients who underwent laparoscopic surgery between January 2019 and December 2020 from the Departments of Gastroenterological Surgery and Hepato-Biliary-Pancreatic Surgery in Okayama University Hospital. To be eligible for inclusion, patients were required to have at least one HbA1c measured within 60 days prior to surgery, which we took as preoperative HbA1c, and at least one glucose measurement after general anesthesia induction immediately prior to surgery, which we took as postinduction glucose. Meanwhile, patients with type 1 diabetes mellitus, who underwent emergency surgery, were aged <20 years, with a surgical duration <60 min, and with a body mass index (BMI) >30 were all excluded from the study. Oral antidiabetic agents and insulin should be stopped on the day of surgery. For patients who underwent reoperative surgery, only the first surgery was included in the study.

### 2.3. Anesthetic Methods

All patients were given propofol, fentanyl with rocuronium during induction of anesthesia. For patients with total intravenous anesthesia, propofol, remifentanil, and rocuronium were used for general anesthesia maintenance. For patients undergoing volatile anesthesia, desflurane or sevoflurane, remifentanil and rocuronium are used for general anesthesia maintenance. The combined use of epidural anesthesia depended on the clinician’s judgment.

### 2.4. Temperature Collection and Management

Basal body temperature was defined as axillary temperature (Terumo Electronic Thermometer; Terumo, Tokyo, Japan) measured on the morning of surgery. Core body temperature was collected after induction, 30 min after surgical incision, 60 min after surgical incision, at the end of the procedure, as well as the minimum temperature from the anesthesia records, using zero-heat-flux (3M Bair Hugger Thermometers; 3M, Maplewood, MN, USA) or bladder thermometers (BARD Silver lubri-sil foley tray with temperature sensor; CR Bard, Murray Hill, NJ, USA) in all enrolled patients. Forced air warming was employed in all patients immediately after induction of anesthesia, and the temperature was set at 38.0 °C and positioned beneath the whole body. The operating room temperature was kept at 23 to 24 °C, with 50 to 60% relative humidity.

### 2.5. Definitions

All preoperative HbA1c measurements were obtained from the electronic health records, and intraoperative glucose and temperature measurements were obtained from the anesthesia information management systems. Diabetes was defined according to a documented history of diabetes from the electronic health records and preoperative anesthesia assessment preoperatively. Based on previous studies [21,22], we considered an HbA1c >6.0% as out-of-range preoperative HbA1c and glucose levels >150 mg/dL or <110 mg/dL as out-of-range postinduction glucose. Furthermore, intraoperative hypothermia was defined as having a core body temperature <36.0 °C [23].

### 2.6. Outcomes

The primary outcome of this study was the incidence of intraoperative hypothermia. Secondary outcome measures incidence and duration of hypothermia, as well as minimum temperature of diabetic patients with different postinduction glucose.

### 2.7. Statistical Analyses

Statistical analyses were performed with the Stata/SE, version 17.0 software (StataCorp, College Station, TX, USA). Data were expressed as medians (25% and 75% quartiles) or frequencies (%), and the level of statistical significance was set at *p* < 0.05. The Shapiro-Wilk W test was applied to check for normal distribution, and data were statistically tested using analysis of variance, Mann-Whitney U test, Chi square test, or Kruskal-Wallis test when appropriate. Dunn’s multiple comparison was applied for post hoc comparisons. A multivariable logistic regression model was also utilized to investigate the association between hypothermia and out-of-range postinduction glucose. Factors with *p* < 0.1 on univariate analysis were included in the multivariate analysis to reveal potential risk factors. Moreover, univariate analyses were performed and compared between patients with and without diabetes. Furthermore, the patients with diabetes were also divided based on their postinduction glucose levels (<110, 110–150, and >150 mg/dL), and the incidence and duration of hypothermia, minimum temperature, and intraoperative temperature changes were compared among these groups.

## 3. Results

### 3.1. Study Population and Patient Characteristics

Of the 821 patients who underwent laparoscopic surgery, 650 fulfilled the study criteria. Among them, 316 were excluded for missing data, resulting in the inclusion of 334 patients in the final analyses. Overall, 200 (59.9%) patients developed hypothermia (Figure 1). Regarding preoperative status, patients who were older and had a lower BMI presented with hypothermia. In contrast, there was no significant difference in the gender ratio, ASA status, history of diabetes and other comorbidities, and HbA1c between hypothermic and normothermic patients. Regarding intraoperative status, the median body temperature was lower in the hypothermic patients than in the normothermic patients at each sampling time point. However, there was no significant difference in anesthesia type, surgery type, surgical duration, fluid input and output, and length of stay between hypothermic and normothermic patients (Table 1, Appendix A). Rapid induction sequence was used in all patients. There was no significant hemodynamic change during the induction.

### 3.2. Hypothermic Patients Presented with a Lower Temperature

In hypothermic patients, the temperature decreased during the surgical process, whereas in normothermic patients, there was no significant change in temperature. At each sampling time point, hypothermic patients had lower temperatures than normothermic patients, and the difference gradually increased with time (Table 2).

### 3.3. Perioperative Glucose Variables Associated with Hypothermia

Postinduction glucose measurements were performed at a median of 42 [34; 54] minutes after the completion of intubation. In univariate analysis, the out-of-range preoperative HbA1c rate was higher in hypothermic patients than in normothermic patients, although this was not statistically significant (*p* = 0.291; Table 2). Meanwhile, postinduction glucose levels were significantly different between hypothermic and normothermic patients (*p* = 0.009; Table 2). Median HbA1c values and postinduction glucose values were similar between hypothermic and normothermic patients (Table 2). However, in multivariate analysis, out-of-range postinduction glucose remained significantly associated with hypothermia, regardless of age ≥65 years, BMI, and ASA status (>150 mg/dL: *p* = 0.045; <110 mg/dL: *p* = 0.015; Table 3).

### 3.4. Out-of-Range Postinduction Glucose in Diabetics Is at a Higher Risk for Hypothermia

Considering only diabetic patients (*n* = 121), the out-of-range postinduction glucose rate was significantly higher in hypothermic patients than in normothermic patients in univariate analysis (*p* = 0.002; Table 4). Similarly, there was no significant difference in the out-of-range postinduction glucose rate of patients without diabetes (*p* = 0.294; Table 4).

On further analysis, in which patients with diabetes were divided into three groups according to postinduction glucose levels, low (<110 mg/dL; *n* = 16), mid (110–150 mg/dL; *n* = 79), and high (>150 mg/dL; *n* = 26), the three groups were significantly different in the incidence of hypothermia (*p* = 0.002; Figure 2A). Additional post hoc analyses showed the mid group had the shortest duration of hypothermia (*p* = 0.007; Figure 2B) and highest minimum temperature (*p* = 0.006; Figure 2C). At each sampling time point, the temperature was always highest in the mid group and lowest in the low group, although there were no statistically significant differences in multivariate analysis of variance (*p* = 0.193; Figure 2D). The temperatures were significantly different between the groups at 60 min after surgical incision (*p* = 0.012; Figure 2D) and at the end of surgery (*p* = 0.038; Figure 2D). In addition, significant differences in the intraoperative temperature changes were also observed between three groups from postinduction to 60 min after surgical incision (*p* = 0.032; Figure 2D), and from 60 min after surgical incision to the end of surgery (*p* = 0.037; Figure 2D).

## 4. Discussion

We conducted this retrospective observational study to assess the perioperative glucose levels of hypothermic and normothermic patients. Although we did not find differences in the comparison of median values, we found that a postinduction glucose >150 mg/dL or <110 mg/dL was a risk factor for intraoperative hypothermia, even after adjusting for covariates. Patients with diabetes who had out-of-range postinduction glucose were also more susceptible to hypothermia than in patients without diabetes. Furthermore, among patients with diabetes, the incidence and duration of hypothermia were significantly higher, and minimum temperature was significantly lower in those who had out-of-range postinduction glucose. Given this, we believe that postinduction glucose is a risk factor for hypothermia, especially in patients with diabetes.

In this study, we assessed preoperative short-term (blood glucose) and long-term (HbA1c) glucose control as risk factors for hypothermia. Preoperative short-term glucose control has been associated with wound complications [24], myocardial injury after noncardiac surgery, and 30-day mortality [25]. Likewise, preoperative long-term glucose control has been associated with postoperative complications and long-term survival [26,27]. There is no consensus on which strategy of perioperative glycemic control is better for improving perioperative complications of patients undergoing surgery [28]. Recent studies reported that intensive glucose control may reduce the risk of infectious complications but also appears to increase the risk of hypoglycaemia [29,30]. Another study indicated that short-term aggressive glucose control was associated with improvements in microvascular function [31]. Therefore, moderate control of perioperative blood glucose levels may be necessary. In our study, we found that preoperative short-term glucose control, which was represented using postinduction glucose, was a risk factor for hypothermia. However, although blood test results are routinely considered on the morning of surgery, a study has suggested that the fasting blood glucose of surgical patients may be influenced by preoperative fasting duration [32]. Therefore, assessing blood glucose levels prior to the start of the procedure may be more predictive, which is why we explored the relationship between postinduction glucose and hypothermia. Moreover, we included the effect of anesthesia in the multivariate logistic regression model, showing that hypothermia was associated with postinduction glucose independent of these parameters.

Although both high and low blood glucose levels can increase the risk of intraoperative hypothermia, the underlying mechanisms may be different. Hypothermia was known to be associated with low blood glucose levels in previous studies [33,34]. Specifically, hypothermia is caused by peripheral vasodilatation and reductions in heat production during hypoglycemia. Peripheral vasodilatation exacerbates the redistribution of heat, resulting in a decreased core body temperature, and reduction in heat production leads to a further decrease in core body temperature [5]. Similarly, hyperglycemia can also have a negative effect on thermoregulation, albeit indirectly. To the best of our knowledge, there has been no study directly correlating hyperglycemia with hypothermia. However, Kenny et al. proposed that hyperglycemia can cause an increase in plasma osmolality, which has been reported to impair vasoconstrictor response to cold [19,35]. Hyperglycemia has also been reported to cause osmotic diuresis and even dehydration without adequate fluid replacement [36]. Interestingly, rehydration and increased urine output in these cases can take away more heat and further reduce the core body temperature. We, therefore, concluded that the postinduction glucose level may not have a direct effect on hypothermia but can indirectly affect the core body temperature by influencing thermoregulation.

Since HbA1c can reflect the mean glucose level for over 2–3 months, elevated HbA1c levels may indicate chronic hyperglycemia. Thus, we chose HbA1c to represent preoperative long-term glucose control in this study. Based on previous studies, we know that the impairment of microvascular function is associated with HbA1c levels, which may make individuals with high HbA1c levels more susceptible to hypothermia [37]. Contrary to our expectations, HbA1c levels in our study were not associated with hypothermia.

On analyzing patients separately based on the presence of diabetes, the same risk factors yielded different results, with out-of-range postinduction glucose being a significant risk factor only in patients with diabetes. Previous studies have noted that hypothermia is more prevalent in patients with diabetes than in non-diabetics [20,38]. This may be explained by the complications of autonomic neuropathy and microvascular disease. One study reported that there was no significant increase in the core body temperature of diabetic patients with autonomic neuropathy under external cooling when compared with diabetic patients without autonomic neuropathy and with non-diabetics [39]. Another study also noted that the active cutaneous vasodilation threshold was higher in patients with diabetes during the entire body-heating process [40].

In this study, age was a risk factor for hypothermia, which was consistent with another prospective cohort study on laparoscopic surgery and studies reporting a 1.8-fold risk for hypothermia in patients ≥70 years [41,42]. The study also showed that patients with a lower BMI were more likely to develop hypothermia, which was consistent with previous studies [41,43]. However, this must be interpreted with caution, as obesity is associated with diabetes and cardiovascular diseases [44]. Regarding ASA, patients with a higher ASA status were at an increased risk for hypothermia, which was consistent with other studies [45,46]. Furthermore, patients with low core body temperatures in our study were at a higher risk for intraoperative hypothermia, which was consistent with two previous randomized controlled trials on preoperative warming [47,48].

The results of our study should be interpreted in the context of its limitations. First, as this was a single-center retrospective study on laparoscopic surgery, our results cannot be applied to all patients and disease entities. Second, although multiple variables were controlled in multivariate analysis, confounding factors may still be involved. Third, we were not able to assess the specific fasting time, which may have an effect on blood glucose. Fourth, we did not routinely examine fasting glucose on the day of surgery, and fasting glucose on the day of surgery may be associated with intraoperative hypothermia. Fifth, the preoperative preparation of the gastrointestinal tract varies among patients due to the type of surgery, which may also have an effect on blood glucose. Sixth, unlike previous reports [49,50], we did not detect an association between the development of hypothermia and increased length of ICU or hospital stay. This may have been due to the difficulty in interpreting this association, as many other confounding variables can account for increased ICU and hospital stay. Lastly, we were unable to demonstrate whether the association between postinduction glucose level and hypothermia was causal or merely an association. Consequently, multicenter prospective studies of perioperative glycemic control and intraoperative hypothermia would be helpful to address these limitations.

## 5. Conclusions

In the present study, we found that postinduction glucose levels may affect intraoperative hypothermia. Postinduction glucose level was shown to be an independent risk factor associated with intraoperative hypothermia. Out-of-range postinduction glucose appeared to have an impact on the development of hypothermia in patients with diabetes, especially those with a postinduction glucose level <110 mg/dL. From the present point of view, this preliminary study offers the possibility of a new randomized controlled trial to assess the effect of perioperative glycemic control on intraoperative hypothermia risk reduction. Further studies will also help to explore the impact of potential mechanisms from biochemical neuropathy on perioperative glycemic control to reduce the incidence of intraoperative hypothermia.

## Figures and Tables

**Figure 1 medicina-59-00395-f001:**
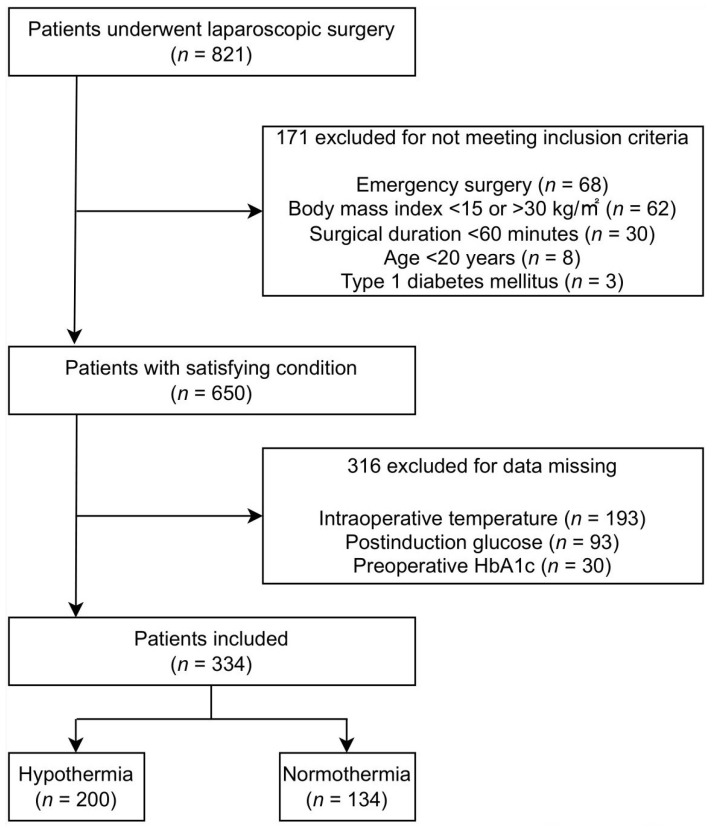
Study consort flow diagram.

**Figure 2 medicina-59-00395-f002:**
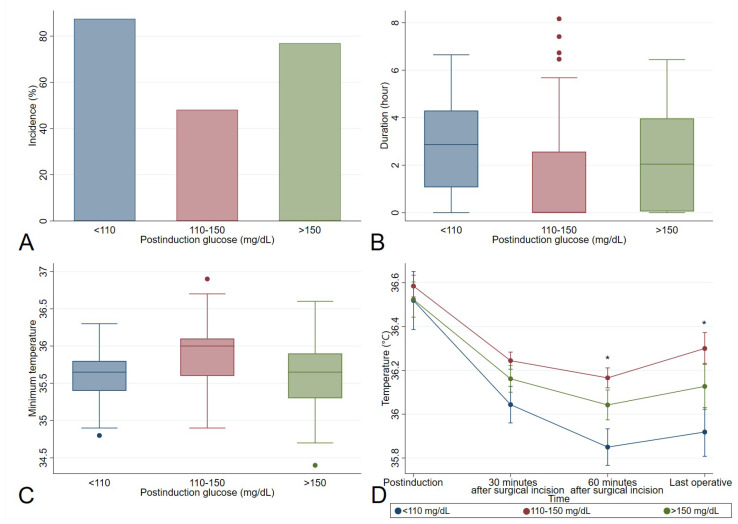
Hypothermia and postinduction glucose in patients with diabetes. (**A**) Incidence of hypothermia in postinduction glucose <110 (87.5%), 110–150 (48.1%), >150 mg/dL (76.9%) patients. The mid group has the lowest incidence of hypothermia. (**B**) Duration of hypothermia in postinduction glucose <110 (2.9 (1.0; 4.4)), 110–150 (0.0 (0.0; 2.6)), >150 (2.0 (0.0; 4.0)) mg/dL patients. The mid group has the shortest duration of hypothermia. (**C**) Minimum temperature in postinduction glucose <110 (35.7 (35.4; 35.8)), 110–150 (36.6 (35.6; 36.1)), >150 (35.7 (35.3; 35.9)) mg/dL patients. The mid group has the highest minimum temperature. (**D**) Core body temperature at each sampling time point in postinduction glucose <110, 110–150, >150 mg/dL patients. Data are shown as means ± SEM. Significant differences between the groups are indicated, *p*-value from Kruskal-Wallis test are: * *p* < 0.05. The mid group had the highest core body temperature at each sampling time point.

**Table 1 medicina-59-00395-t001:** Patient characteristics for hypothermic and normothermic patients.

	Hypothermic (*n* = 200)	Normothermic (*n* = 134)	*p* Value
Age (years)	70 [65; 77]	68 [56; 74]	0.007
Age ≥65 years *n* (%)	151 (65.9)	78 (34.1)	0.001
Male *n* (%)	122 (61.0)	85 (63.4)	0.653
BMI (kg/m^2^)	22.3 [20.1; 24.5]	23.0 [21.5; 25.3]	0.003
ASA-Score *n* (%)			0.070
I	35 (17.5)	35 (26.1)	
II	129 (64.5)	84 (62.7)	
>II	36 (18.0)	15 (11.2)	
Diabetes *n* (%)	72 (36.0)	49 (36.6)	0.916
Intraoperative parameters			
Total intravenous/volatile anesthesia *n*	62/138	34/100	0.265
Epidural-general anesthesia/general anesthesia *n*	131/69	98/36	0.141
Duration of surgery (hour)	4.8 [3.7; 6.5]	4.8 [3.6; 7.3]	0.963
Intravenous fluid administration (mL)	2516 [1906; 3161]	2398 [1728; 3335]	0.605
Estimated blood loss and urine output (mL)	453 [266; 775]	415 [238; 753]	0.557
Intraoperative insulin dose (IU) (*n* = 8)	(*n* = 4) 6.4 [1.6; 9.6]	(*n* = 4) 6.0 [2.2; 9.1]	0.999
Intraoperative glucose dose (g)	13.3 [10.0; 15.0]	14.0 [10.0; 15.0]	0.906
Outcome parameters			
Postoperative hospital length of stay (days)	13 [10; 19]	12 [10; 22]	0.971
ICU length of stay (days)	(*n* = 49) 4 [2; 6]	(*n* = 36) 5 [2; 5]	0.757

Data are shown as median with quartiles (25%; 75%) or as frequencies *n* (%). BMI: Body Mass Index; ASA: American Society of Anesthesiologists; ICU: Intensive Care Unit.

**Table 2 medicina-59-00395-t002:** Perioperative glucose and body temperature variables for hypothermic and normothermic patients.

	Hypothermic (*n* = 200)	Normothermic (*n* = 134)	*p* Value
Perioperative glucose variables			
HbA1c (%)	5.9 [5.5; 6.4]	5.8 [5.4; 6.2]	0.182
HbA1c >6% *n* (%)	77 (38.5)	44 (32.8)	0.291
Postinduction glucose (mg/dL)	119 [106; 138]	125 [112; 137]	0.063
Postinduction glucose levels *n* (%)			0.009
Postinduction glucose <110 mg/dL	60 (30.0)	24 (17.9)	
Postinduction glucose 110–150 mg/dL	112 (56.0)	97 (72.4)	
Postinduction glucose >150 mg/dL	28 (14.0)	13 (9.7)	
Perioperative body temperature variables			
Basal body temperature (°C)	36.4 [36.2; 36.6]	36.4 [36.2; 36.5]	0.949
Basal body temperature <36 °C *n* (%)	23 (11.5)	13 (9.7)	0.603
Postinduction temperature (°C)	36.4 [36.1; 36.7]	36.7 [36.6; 37.0]	<0.001
30 min after surgical incision temperature (°C)	36.0 [35.8; 36.2]	36.5 [36.3; 36.7]	<0.001
60 min after surgical incision temperature (°C)	35.9 [35.7; 36.1]	36.5 [36.3; 36.7]	<0.001
Last operative temperature (°C)	35.9 [35.6; 36.2]	36.6 [36.4; 37.0]	<0.001

Data are shown as median with quartiles (25%; 75%) or as frequencies *n* (%). HbA1c: Hemoglobin A1c.

**Table 3 medicina-59-00395-t003:** Multivariable logistic regression model analyzing predictors for hypothermia.

	OR (95% CI)	*p* Value
Age ≥65 years	2.03 (1.22, 3.39)	0.007
BMI	0.87 (0.80, 0.95)	0.001
ASA-Score	1.36 (0.90, 2.06)	0.148
Postinduction glucose levels		
Postinduction glucose 110–150 mg/dL *		
Postinduction glucose <110 mg/dL	2.02 (1.15, 3.55)	0.015
Postinduction glucose >150 mg/dL	2.17 (1.02, 4.61)	0.045

OR: Odds Ratio; 95% CI: 95% Confidence Interval; BMI: Body Mass Index; ASA: American Society of Anesthesiologists; *: Reference.

**Table 4 medicina-59-00395-t004:** Patient characteristics in patients with and without diabetes.

	Diabetic (*n* = 121)			Non-Diabetic (*n* = 213)		
	Hypothermic (*n* = 72)	Normothermic (*n* = 49)	*p* Value	Hypothermic (*n* = 128)	Normothermic (*n* = 85)	*p* Value
Age ≥65 years *n* (%)	61 (84.7)	36 (73.5)	0.128	90 (70.3)	42 (49.4)	0.002
BMI (kg/m^2^)	22.6 [20.5; 24.5]	23.2 [21.2; 25.2]	0.183	21.9 [19.9; 24.5]	22.9 [21.5; 25.3]	0.007
ASA-Score *n* (%)			0.162			0.125
I	10 (13.9)	8 (16.3)		25 (19.5)	27 (31.8)	
II	41 (56.9)	34 (69.4)		88 (68.8)	50 (58.8)	
>II	21 (29.2)	7 (14.3)		15 (11.7)	8 (9.4)	
Basal body temperature (°C)	36.4 [36.1; 36.6]	36.4 [36.1; 36.5]	0.475	36.4 [36.2; 36.5]	36.4 [36.2; 36.6]	0.619
Preoperative HbA1c >6% *n* (%)	49 (68.1)	28 (57.1)	0.221	28 (21.9)	16 (18.8)	0.590
Postinduction glucose levels *n* (%)			0.002			0.294
Postinduction glucose <110 mg/dL	14 (19.4)	2 (4.1)		46 (35.9)	22 (25.9)	
Postinduction glucose 110–150 mg/dL	38 (52.8)	41 (83.7)		74 (57.8)	56 (65.9)	
Postinduction glucose >150 mg/dL	20 (27.8)	6 (12.2)		8 (6.3)	7 (8.2)	
Intravenous fluid administration (mL)	2683 [1959; 3429]	2450 [1702; 3216]	0.156	2458 [1894; 3098]	2394 [1772; 3690]	0.740
Estimated blood loss and urine output (mL)	493 [275; 949]	305 [228; 715]	0.028	438 [223; 708]	465 [273; 775]	0.326

Data are shown as median with quartiles (25%; 75%) or as frequencies *n* (%). BMI: Body Mass Index; ASA: American Society of Anesthesiologists; HbA1c: Hemoglobin A1c.

## Data Availability

The data that support the findings of this study are available from the corresponding author upon reasonable request.

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
