# Peer review of "The Effect of Postinduction Blood Glucose on Intraoperative Hypothermia"

_medicina, 2023, doi:10.3390/medicina59020395_

Round 1

Reviewer 1 Report

This is an retrospective and observational study conducted to investigate the association between intraoperative hypothermia and blood glucose levels. Like all the observational study, it is not expected to demonstrate causality but only association between variables. The sample size was adequate and the statistical analyses was also appropriate.  

1. Review the english grammar at lines 122 and 123.

2. Table 1 can be improved to a better format and highlight "ASA-Score", "perioperative glucose levels", postinduction glucose levels, etc.

3. Table 2. Highlight "postinduction glucose levels" and change reference values to table footnote.

4. It would have been interesting to report on the fasting blood glucose values of patients and their association results.

5. Although this study has found an important association, it is necessary to  include more information about the different types of surgery and the types of anesthetics used. 

Author Response

Reviewer #1

Thank you very much for your valuable comments and suggestions on our manuscript entitled “The effect of postinduction blood glucose on intraoperative hypothermia”. In reply to your comments, we modified the manuscript one-by-one listed below. We believe that the manuscript has been improved substantially. We sincerely appreciate your contribution to this revision.

This is an retrospective and observational study conducted to investigate the association between intraoperative hypothermia and blood glucose levels. Like all the observational study, it is not expected to demonstrate causality but only association between variables. The sample size was adequate and the statistical analyses was also appropriate. 

  1. Review the english grammar at lines 122 and 123.

RESPONSE:

Thank you very much for your indication. According to your indication, we have corrected the English grammar as follows. “Regarding intraoperative status, the median body temperature was lower in the hypo-thermic patients than in the normothermic patients at each sampling time point.”

  1. Table 1 can be improved to a better format and highlight "ASA-Score", "perioperative glucose levels", postinduction glucose levels, etc.

RESPONSE:

Thank you for your indication. We have optimized the presentation of the table and split the table in our revised version.

  1. Table 2. Highlight "postinduction glucose levels" and change reference values to table footnote.

RESPONSE:

Thank you very much for your indication. According to your indication, we have optimized the presentation of the table and we have used table footnote to make the table simpler and clearer.

  1. It would have been interesting to report on the fasting blood glucose values of patients and their association results.

RESPONSE:

Thank you very much for your indication. According to your indication, we counted the preoperative fasting glucose values of all patients, but unfortunately, as this retrospective study, only 11 patients were performed with preoperative fasting glucose values, and this test is not used as a routine test in our hospital. We have included this as a limitation in the discussion section as follows. “We did not routinely examine fasting glucose on the day of surgery, and fasting glucose on the day of surgery may be associated with intraoperative hypothermia.” We hope to refine this in future prospective studies.

  1. Although this study has found an important association, it is necessary to include more information about the different types of surgery and the types of anesthetics used.

RESPONSE:

Thank you for your indication. According to your indication, we have supplemented the supplemental table with content that may be of interest to you, also we added Anesthetic methods to the methods section as follows. “All patients were given propofol, fentanyl with rocuronium during induction of anaesthesia. For patients with total intravenous anesthesia, propofol, remifentanil, and rocuronium were used for general anesthesia maintenance. For patients undergoing volatile anesthesia, desflurane or sevoflurane, remifentanil and rocuronium are used for general anesthesia maintenance. The combined use of epidural anaesthesia was depending on the clinician's judgment.” Regarding the types of procedures, in laparoscopic surgery we usually classify them into the four categories in the table. However, we were unable to find a statistical difference. This may be further detailed in a future study.

Reviewer 2 Report

The study “The effect of postinduction blood glucose on intraoperative hypothermia” gives highlights to some very important homeostasis of the body, such as relation between blood glucose levels and body temperature homeostasis. Anyway, the idea is old and the novelty of this study should be properly checked. The scientific background and deeper analysis of this problem should be described more in detail, such as the rationale for the study. The literature review is not strong enough to provide research gaps for this research work. Introduction section is too short, and should be a little bit empowered according to latest biochemical neuropathy mechanisms. It is obvious the quality of the manuscript can meet the standards of Medicina Journal after major revisions. My suggestion is major revision.

Specific comments:

1.     Authors are advised to revise keywords, and to give some more specific one.

2.     Introduction section is too short, and poorly describes some very well-known mechanism. Authors are suggested to empower the Introduction section according to latest knowledge from Biochemistry related to diabetic neuropathy (Lines 39-45). Authors are suggested to use following references for this:  Clinical use of an analysis of oxidative stress and IL-6 as the promoters of diabetic polyneuropathy; Medicinski Glasnik, 2021; 18(1):1–6. Diabetes Research and Clinical Practice, 2022; 186: 109682. https://doi.org/10.1016/j.diabres.2022.109682.

3.     Results section needs major revision in presentation. Authors are asked to make some more graphs and figures, for better interpretation. The tables seems to congested, and confusing in some points.

4.     Conslusion section should be revised, according to following instructions: Key values from results; Major findings and contribution; As well as limitations of the study if there is any; Possible future work; Make it up to 250-300 words.

5.     References should be updated by the new one, as many references are older than 10 years; Cite more references from the last 4-5 years 2018-2022 (50-70%), and empower the Introduction and Discussion section by the latest one, as suggested.

6.     Number of references should be increased.

Author Response

Reviewer #2

Thank you very much for your valuable comments and suggestions on our manuscript entitled “The effect of postinduction blood glucose on intraoperative hypothermia”. In reply to your comments, we modified the manuscript one-by-one listed below. We believe that the manuscript has been improved substantially. We sincerely appreciate your contribution to this revision.

The study “The effect of postinduction blood glucose on intraoperative hypothermia” gives highlights to some very important homeostasis of the body, such as relation between blood glucose levels and body temperature homeostasis. Anyway, the idea is old and the novelty of this study should be properly checked. The scientific background and deeper analysis of this problem should be described more in detail, such as the rationale for the study. The literature review is not strong enough to provide research gaps for this research work. Introduction section is too short, and should be a little bit empowered according to latest biochemical neuropathy mechanisms. It is obvious the quality of the manuscript can meet the standards of Medicina Journal after major revisions. My suggestion is major revision.

Specific comments:

  1. Authors are advised to revise keywords, and to give some more specific one.

RESPONSE:

Thank you very much for your indication. According to your indication, we have updated the keywords and made them as compliant as possible with the MeSH specification.

  1. Introduction section is too short, and poorly describes some very well-known mechanism. Authors are suggested to empower the Introduction section according to latest knowledge from Biochemistry related to diabetic neuropathy (Lines 39-45). Authors are suggested to use following references for this: Clinical use of an analysis of oxidative stress and IL-6 as the promoters of diabetic polyneuropathy; Medicinski Glasnik, 2021; 18(1):1–6. Diabetes Research and Clinical Practice, 2022; 186: 109682. https://doi.org/10.1016/j.diabres.2022.109682.

RESPONSE:

Thank you very much for your indication. According to your indication, we have cited some of the latest knowledge of the biochemistry associated with diabetic neuropathy, including the references you provided. Thank you for the point of view of internal medicine, and in the process of further refining the description of the mechanism, we have also identified some inflammatory factors in relation to hypothermia. We have updated the introduction section as follows. “Clinically, short- and long-term glucose control are also associated with postoperative outcomes [10,11]. Although, current evidence shows that poor glycemic control can lead to peripheral neuropathy and microvascular dysfunction [12], the mechanism of pathophysiology is very complex. Commonly used explanation is oxidative stress [13]. Inflammatory cytokines have the main control in these processes because of their inflammatory promoting activity [14,15], which in turn may related to intraoperative hypothermia [16,17]. However, the mechanism of how blood glucose affects thermoregulation remains unclear. Further information regarding this may be gleaned from studies in patients with diabetes. Due to diabetic autonomic neuropathy and diabetic microvascular disease, these patients appear to have an impaired vascular responsiveness to cold [18], resulting in a reduced ability to maintain their core temperature during cold exposure [19]. Therefore, active temperature management and moderate preoperative glycemic control could be important to enhance patient safety and satisfaction.”

  1. Results section needs major revision in presentation. Authors are asked to make some more graphs and figures, for better interpretation. The tables seems to congested, and confusing in some points.

RESPONSE:

Thank you for your indication. According to your indication, we have optimized the presentation of the table and split the table in our revised version. In the revised version, the more noteworthy values are presented in Table 2. Regarding not using more figures, we think the presentation in table form is more suitable for the type of our results.

  1. Conslusion section should be revised, according to following instructions: Key values from results; Major findings and contribution; As well as limitations of the study if there is any; Possible future work; Make it up to 250-300 words.

RESPONSE:

Thank you very much for your indication. According to your indication, we have updated the conclusions section as follows. “In the present study, we found that postinduction glucose levels may affect intraoperative hypothermia. Postinduction glucose level was showed to be an independent risk factor associated with intraoperative hypothermia. Out-of-range postinduction glucose appeared to have an impact on the development of hypothermia in patients with diabetes, especially those with a postinduction glucose level <110 mg/dL. From the present point of view, this preliminary study offers the possibility of a new randomized controlled trial to access the effect of perioperative glycemic control on intraoperative hypothermia risk reduction. Further studies will also help to explore the impact of potential mechanisms from biochemical neuropathy on perioperative glycemic control to reduce the incidence of intraoperative hypothermia.”

  1. References should be updated by the new one, as many references are older than 10 years; Cite more references from the last 4-5 years 2018-2022 (50-70%), and empower the Introduction and Discussion section by the latest one, as suggested.

RESPONSE:

Thank you for your indication. According to your indication, we have updated many citations, especially some articles from recent years, and these new both biochemical and clinical studies are of great help to us to better illustrate our points.

  1. Number of references should be increased.

RESPONSE:

Thank you for your indication. According to your indication, we have added a considerable number of references and used the most recent ones as possible.

Round 2

Reviewer 2 Report

The authors have addressed most of the comments; they have also tried to make changes according to the reviewers’ suggestions. After revisions, the quality of the manuscript has been adequately enhanced. Therefore, the manuscript could be considered for publication in the Journal.